# Opposing neural processing modes alternate rhythmically during sustained auditory attention
Florian H. Kasten [1,2,3] ✉, Quentin Busson [3] & Benedikt Zoefel [2,3] ✉

During continuous tasks, humans show spontaneous fluctuations in performance, putatively caused by varying attentional resources allocated to process external information. If neural resources are used to process other, presumably "internal" information, sensory input can be missed and explain an apparent dichotomy of "internal" versus "external" attention. In the current study, we extract presumed neural signatures of these attentional modes in human electroencephalography (EEG): neural entrainment and α-oscillations (~10-Hz), linked to the processing and suppression of sensory information, respectively. We test whether they exhibit structured fluctuations over time, while listeners attend to an ecologically relevant stimulus, like speech, and complete a task that requires full and continuous attention. Results show an antagonistic relation between neural entrainment to speech and spontaneous α-oscillations in two distinct brain networks—one specialized in the processing of external information, the other reminiscent of the dorsal attention network. These opposing neural modes undergo slow, periodic fluctuations around ~0.07 Hz and are related to the detection of auditory targets. Our study might have tapped into a general attentional mechanism that is conserved across species and has important implications for situations in which sustained attention to sensory information is critical.

The ability to sustain attention is crucial for many activities of everyday life, yet it is surprisingly difficult to achieve[1,2]. Lapses in attention are common even in healthy populations, can have negative downstream effects on cognition[3] and lead to human error, sometimes with major consequences[4–6]. A variety of neurological and psychiatric disorders are characterized or accompanied by a decreased ability to maintain sustained attention[7,8]. Understanding the neural processes that give rise to dynamic fluctuations in attention is therefore critical.

It has been proposed that these fluctuations arise due to varying amounts of attentional resources allocated to the processing of external information. When sensory input is ignored, neural resources might be available for other processes unrelated to external information (such as memory consolidation or internal thought), leading to a dichotomy of "internal" versus "external" attention[9,10]. Each of these opposing attentional modes might have its own signature, including specific patterns of neural connectivity[1,2] and neural oscillations[10–12]. Prominently, α-oscillations (~10-Hz) have been linked to suppression of sensory information[13,14] and attentional (de-)selection[15–18], and might therefore correspond to a state in which

input is prone to be ignored. When listeners attend to rhythmic sequences, neural activity synchronizes to the stimulus rhythm, an effect often termed neural entrainment[19–21]. As a potential counterpart to α-oscillations, neural entrainment is therefore a prime example for a marker of active stimulus processing[20,22,23]. However, neural entrainment and α-oscillations are often studied in isolation, and the few efforts to relate them[24–26] have used trial-based paradigms that, due to their frequent interruptions of stimulus presentation, cannot capture fluctuations in sustained attention.

One of the rare studies that did consider both of these neural markers during sustained attention measured them in the auditory cortex of macaque monkeys[10]. This work supported the notion that internal attention is characterized by strong α-oscillations and reduced sensitivity to external information[10]. When subjects listened to rhythmic tone sequences, neural activity synchronized to the stimulus rhythm[19,25,26]. At certain times, however, neural entrainment was reduced and α-oscillations dominated processing in auditory cortex. When α-oscillations prevailed, they rhythmically modulated neuronal firing and reduced neural and behavioral responses to stimulus input. Fundamental for our study, these bouts of α-oscillations (i.e.,

[1]Department for Cognitive, Affective, Behavioral Neuroscience with Focus Neurostimulation, Institute of Psychology, University of Trier, Trier, Germany. [2]Centre de Recherche Cerveau & Cognition, CNRS, Toulouse, France. [3]Université Toulouse III Paul Sabatier, Toulouse, France. ✉e-mail: florian.kasten@uni-trier.de; benedikt.zoefel@cnrs.fr

internal attention) occurred regularly and alternated with periods of strong entrainment to sound (i.e., external attention), at an inherent rhythm of ~0.06 Hz (i.e., ~16 s)[10].

The identification of rhythmicity in attentional states and their neural counterparts has important implications for future research. It could be leveraged in the design of critical systems technology, educational environments or in the design of interventional approaches for situations where sustained attention is critical. However, it remained elusive if humans possess equivalent neural signatures of attentional modes, and whether they exhibit any temporal regularity. It also remained unclear if regular attentional lapses occur during the processing of ecologically relevant stimuli. Human speech requires integration of information over time to be optimally perceived[27]. It therefore needs full and continuous attention, and regular attentional lapses seem particularly harmful for speech processing. Finally, the network structure governing the hypothesized modes in auditory sustained attention remained unaddressed thus far.

In the current study, we recorded electroencephalographic (EEG) data in humans and tested for regular fluctuations in attentional modes while participants paid sustained attention to rhythmic speech sounds. We hypothesized that, similar to the aforementioned findings in non-human primates[10], neural entrainment to speech and spontaneous α-oscillations show rhythmic fluctuations at ultra-slow frequencies close to ~0.06 Hz (0.02 Hz–0.2 Hz) and that these fluctuations show an antagonistic relationship (i.e., are coupled in anti-phase). Given its strong environmental rhythms[28], we here chose to focus on the auditory modality. However, our findings may pave the way for investigations into rhythmic attentional fluctuations across sensory modalities.

## Results
### Overview
We recorded EEG from $n = 23$ participants while they listened to 5-min streams of rhythmic, monosyllabic, French words presented at a rate of 3-Hz (Fig. 1a). Depending on the experimental block, participants were instructed to keep their eyes open or closed, respectively. They were asked to identify words that were presented off-rhythm (i.e., shifted by 80-ms relative to the 3-Hz rhythm). On average, participants detected 41.22% (±SD: 13.80) of targets during the eyes-open and 42.96% (±SD: 15.56) during eyes-closed (Fig. 1b) conditions. The proportion of false alarms was low relative to the large number of non-target words (eyes-open: 0.91% ± SD: 1.06, eyes-closed: 0.83% ± SD: 1.00, Fig. 1c). There was no difference in hits or false alarms between eyes-closed and eyes-open conditions (dependent samples $t$-test, hits: $t_{22} = -0.94$, $p = 0.35$, FA: $t_{22} = 1.11$, $p = 0.27$), nor was there a difference in reaction times ($t_{22} = 1.62$, $p = 0.12$: $M_{eyes-open} = 777$ ms ± SD: 122; $M_{eyes-closed} = 738$ ms ± SD: 96, Fig. 1d).

We used standard spectral analysis methods to extract spontaneous α-oscillations, and inter-trial coherence (ITC)[10,23] at 3 Hz to quantify auditory entrainment (see "Materials and methods"). Both showed EEG topographies consistent with the literature[23,29] (Fig. 1e, f). α-oscillations showed a dominant occipito-parietal topography with a prominent increase in power when participants closed their eyes[29] (Fig. 1e). Auditory entrainment was dominant in fronto-central sensors with peaks in the ITC spectrum at the 3 Hz stimulus rate and its harmonic frequencies[19,23,30] (Fig. 1f).

### α-Oscillations and entrainment show slow fluctuations at similar time scales
Adapting an approach from Lakatos et al.[10] to human EEG, we traced the evolution of spontaneous α-power and auditory entrainment during the task (Fig. 1a). We used a sliding window approach to quantify how both of these measures change over time (see "Materials and methods").

We found that both α-oscillations and neural entrainment exhibit slow, regular fluctuations (Fig. 2a). The dominant frequency in these fluctuations, which we revealed as 0.07 Hz (~14 s, $M_{alpha} = 0.0713$ Hz ± SD = 0.0126, $M_{ITC} = 0.0710$ Hz ± SD = 0.0116), was strikingly similar to that reported in non-human primates[10]. While the topographical distribution of entrainment fluctuations (Fig. 2b) resembled that of entrainment itself (Fig. 1e), this

was not the case for α-oscillations. This result implies that the observed fluctuations in α-power (Fig. 2b) might be indeed linked to auditory attentional processing, in contrast to the distribution of α-power that is generally dominated by the visual system (Fig. 1d).

α-oscillations and entrainment did not only fluctuate at similar time scales on the group level (Fig. 2c), but also within individuals: on average, the individual peak frequency for α-power fluctuations did not differ from that for neural entrainment (dependent samples $t$-test: $t_{22} = 0.08$, $p = 0.93$; $M_{|alpha - ITC|} = 0.015$ Hz ± SD = 0.0112 Hz). Together, we found that α-oscillations and neural entrainment exhibit similar slow, regular fluctuations.

### Anti-phasic relation between α-power and entrainment fluctuations
We next assessed if the rhythmic fluctuations of α-oscillations and auditory entrainment are coupled. If the two reflect opposing processing modes, they should show an anti-phase relationship: When α-oscillations are strong, neural entrainment should be reduced, and vice versa. To this end, we computed the average phase difference between α-power and entrainment fluctuations for each EEG channel. Our analysis revealed a significant coupling (i.e., consistent phase relation) between α-power and entrainment across subjects within a cluster of fronto-central channels (cluster-based Rayleigh's test; $p_{cluster} = 0.037$, Fig. 2d, see Supplementary Table 1 for an overview of channels within the cluster). A circular one-sample test yielded a significant deviation of the average phase difference from zero ($p < 0.001$). This average phase difference was close to anti-phase ($M_{angle} = -3.04$ rad), and with the 99% confidence interval for the sample mean including ± π ($CI_{99} = 2.39$, $-2.18$, Fig. 2f). Supplementary Fig. S1a provides an overview of the phase distribution in each EEG channel. Importantly, the anti-phasic relation of the two signals was evident on single subject level. Fourteen out of 23 participants show a significant coupling within the identified cluster. For 15 out of 23 participants, the average angle between α-power and entrainment fluctuations significantly differed from 0 (Supplementary Fig. S1b). We could not identify reliable differences in oscillatory features or other variables that can distinguish these 15 participants from the others (Supplementary Fig. S2 and Supplementary Note 1).

For results described above and shown in Fig. 2d, we contrasted α-power and entrainment from the same EEG channels. We next tested whether different channel combinations produce similar results. The topographical distribution of auditory entrainment in the EEG is well established and was reproduced in our results (Fig. 2b, d). However, α-oscillations are typically dominated by vision and their topographical pattern was more difficult to predict in our case. We therefore assessed, separately for each channel, whether α-power in this channel is coupled with neural entrainment in the frontal channel cluster shown in Fig. 2d. The analysis revealed a more distributed cluster of fronto-central and parietal channels in which α-power fluctuations are coupled to auditory entrainment (random permutation cluster Rayleigh test: $p_{cluster} = 0.042$, Fig. 2e, see Supplementary Table 1 for an overview of channels within the cluster). Again, the difference between entrainment and α-power fluctuations was close to anti-phase within this cluster ($M_{angle} = 3.13$ rad, circular one-sample test against angle of zero: $p < 0.01$) with the 99% CI for the sample mean including ± π ($CI_{99} = 2.16$, $-2.18$, Fig. 2g). Figure 2h depicts an exemplary time course of α-power and entrainment fluctuations. An example for each participant is shown in Supplementary Fig. S1c. In control analyses, we ruled out that the fluctuations observed in the α-band are driven by harmonic entrainment at frequencies in the α-band (Supplementary Fig. S3). Together, we found an anti-phase relation between the slow fluctuations in α-power and neural entrainment, as predicted from two opposing neural processes[9,12].

### Anti-phase relation between entrainment and α-oscillations is state dependent
When participants were instructed to close their eyes during the task, the 0.07 Hz peaks became less pronounced, compared to the eyes-open

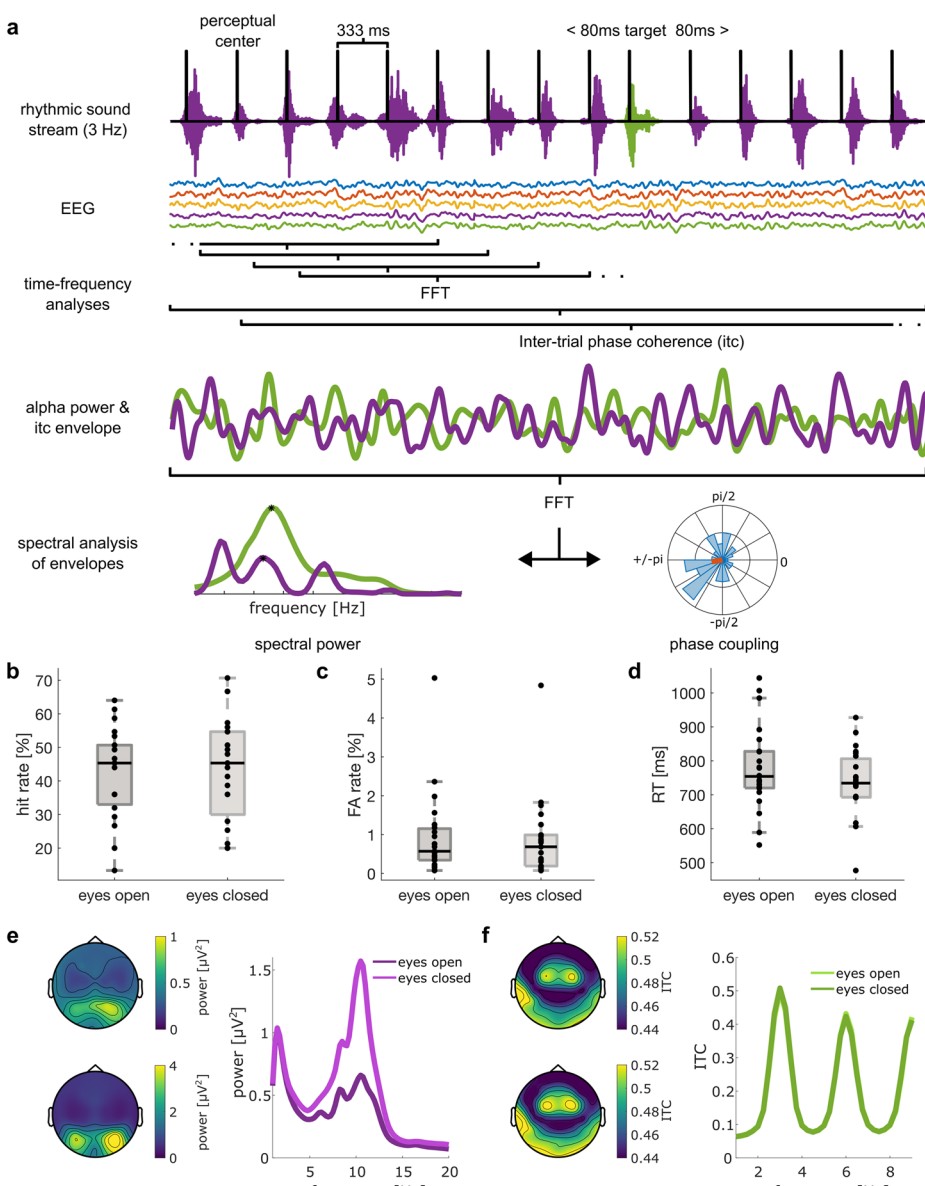

**Fig. 1 | Experimental design and analysis. a** Participants listened to continuous 5-min streams of rhythmic, monosyllabic French words presented at a rate of 3 Hz (top). Spectral analysis was performed on 2 s EEG segments centered on the perceptual center of each word. Fifteen adjacent segments (~5 s window) were integrated in a sliding window approach to compute inter-trial coherence (ITC) over time. ITC at 3 Hz and power in the α-band (8–12 Hz) were extracted and treated as new time-series (4th row). The two time-series were submitted to another spectral analysis to assess slow, rhythmic fluctuations of α-oscillations and auditory entrainment. We identified prominent spectral peaks in both spectra and assessed their coupling and phasic relation (bottom row). **b, c** Proportion of hits and false alarms in eyes-open and eyes-closed conditions. **d** Reaction times for hits. **e** Topography of α-power in eyes-open (top) and eyes-closed (bottom) conditions. Spectra on the right have been extracted from channel Pz and averaged across subjects. **f** Same as (**e**), but for ITC (spectra are shown for channel Fz). Both α-power and ITC spectra and topographies are consistent with previous reports in the literature. Boxplots depict the median of the data ± interquartile range. Whiskers indicate the range of the data.

condition (Fig. 2i–l). Further, we did not find evidence for a significant coupling ($p_{cluster} > 0.68$, Fig. 2k) or an anti-phase relationship (circular one-sample test against angle of zero: $p > 0.05$, $M_{angle} = 2.22$ rad, Fig. 2l) between α-oscillations and entrainment when the eyes were closed.

## Slow rhythmic fluctuations cannot be explained by stimulus properties

Although time intervals between targets were selected from a wide range and therefore irregular, their average (25 targets in 300 s) resembled the period of the slow neural fluctuations reported. To rule out that some regularity in target presentation explains the slow rhythmicity in α-power or auditory entrainment, we computed the distribution of time intervals between targets

(Fig. 3a), hits (Fig. 3b), and misses (Fig. 3c). None of these distributions shows a bias for 14 s (which would correspond to the 0.07 Hz rhythm found in α-power and ITC). Moreover, we found that individual frequencies for slow changes in α-power and ITC are uncorrelated with corresponding intervals between targets (Fig. 3d, g), hits (Fig. 3e, h), or misses (Fig. 3f, i). It therefore seems unlikely that the rate of target presentation gave rise to the observed fluctuations in the EEG.

## Opposing attentional modes emerge from interactions of distinct cortical networks

To reveal the neural sources of opposing attentional modes, we source localized the slow rhythmic fluctuations in neural entrainment (Fig. 4a) and α-power

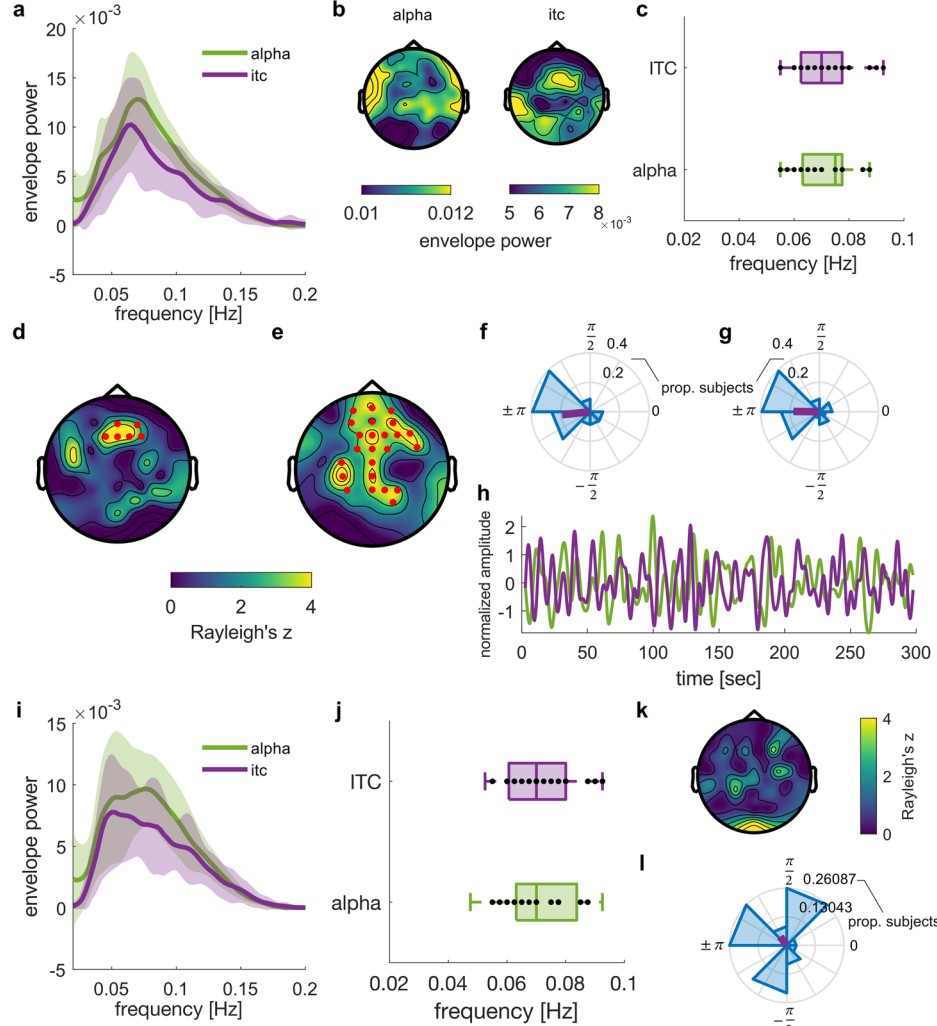

**Fig. 2 | α-power and entrainment exhibit slow anti-phase fluctuations.**
**a–h** Results for eyes-open condition. **a** Envelope spectra of α-power and entrainment to rhythmic speech show a peak around 0.07 Hz (shown for electrode Fz). **b** Topography of the 0.07 Hz peak shown in (**a**). **c** Distribution of individual peak frequencies from the spectra shown in (**a**). **d** Coupling between α-power and entrainment fluctuations around 0.07 Hz when both were extracted from the same channel. Channels showing a significant non-uniform distribution of phase differences are highlighted in red, topography indicates the underlying z-statistic.

**e** Channels showing significant coupling between α-power fluctuations per channel with entrainment in the frontal cluster (**d**). **f** Distributions of phases in channels showing significant α-power vs. entrainment coupling (cluster shown in **d**). α-power and entrainment to speech are coupled in anti-phase. **g** α-power fluctuations in cluster shown in (**e**) are coupled in anti-phase. **h** Exemplary time-course of α-power and entrainment fluctuations at electrode Fz. **i–l** Same as (**a, c, d, g**), but for eyes-closed condition. Shaded areas depict standard deviation. Boxplots depict the median of the data ± interquartile range. Whiskers indicate the range of the data.

(Fig. 4b) in the eyes-open condition, using a frequency domain beamformer[31] at the individual peak frequency of α-power fluctuations used in the previous analyses (Fig. 2c). For both of these, we observed activity in regions associated with auditory and speech processing including Superior Temporal Gyrus (STG), as well as left Pre-Central Gyrus. For entrainment (Fig. 4a), we found an additional involvement of left Inferior Frontal Gyrus (IFG) as well as the right Inferior Parietal Lobe, Angular Gyrus and parts of posterior STG. For α-power (Fig. 4b), we found additional activity in left Post-Central Gyrus, Superior Parietal Lobe as well as Posterior Cingulate and Occipital Cortex.

We then quantified coupling between neural entrainment and α-power as a phase relation between the two that is consistent across participants. This approach was similar to the previous sensor-level analysis, only that it was applied to different combinations of 360 cortical regions-of-interest (ROIs), parcellated on the source level according to the HCP-MMP1 atlas of the human connectome project[32]. We found significant coupling between slow entrainment changes bilaterally in ROIs in temporal cortex and left Inferior Frontal Gyrus, and equivalent changes in α-power in bilateral Posterior Cingulate Cortex, Superior Parietal Lobe and right Inferior Frontal Gyrus (permutation cluster Rayleigh test with three significant clusters:

$p_{cluster1} = 0.0026$, $p_{cluster2} = 0.004$, $p_{cluster3} = 0.006$; Fig. 4c). Table 1 provides an overview of ROIs within the significant clusters, Fig. 4d depicts coupling for ROIs sorted into 44 higher order cortical structures according to the HCP-MMP1 atlas. Again, the coupling was driven by a phase relation between ROIs that was close to anti-phase (Fig. 4e). Whereas structures showing coupled fluctuations in entrainment are typically associated with processing of speech[33–36] (and therefore may reflect attention to sensory input), those with corresponding α-fluctuations involve regions associated with fronto-parietal attention networks[37,38], but also the default mode network[38,39]. Together, these results imply that fluctuations in opposing attentional modes may stem from an interaction between distinct cortical networks, one associated with active processing of external stimuli (i.e., the speech network), and the other concerned with up- and down-regulation of processing resources via inhibitory α-oscillations.

## Neural signatures of attentional mode differ between detected and missed target stimuli

Thus far we have reported a systematic coupling between slow fluctuations of α-power and auditory entrainment to rhythmic speech. Periods

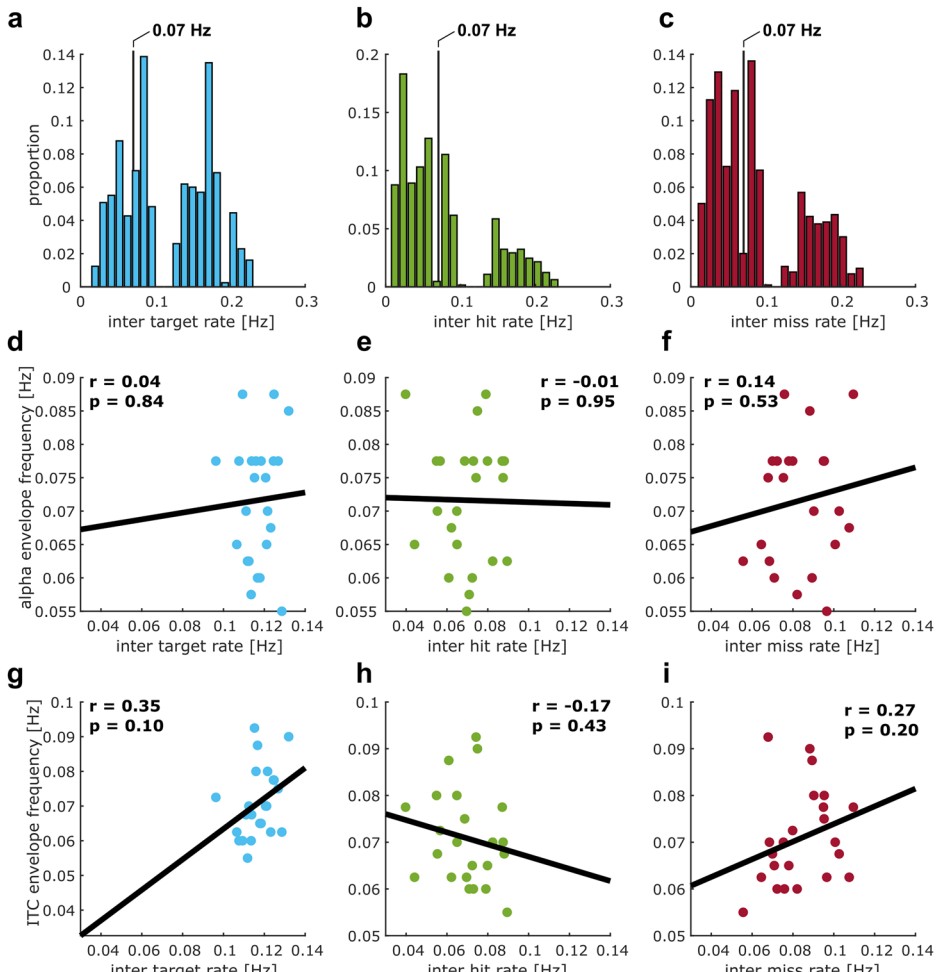

**Fig. 3 | Control analysis.** Distribution of time intervals between targets (**a**), hits (**b**), and misses (**c**), pooled across subjects for the eyes-open condition. Intervals were converted to rate (in Hz) for comparability with neural fluctuations. Vertical line indicates 0.07 Hz. **d–i** Correlations between individual frequencies for slow fluctuations in alpha power (middle row) or ITC (bottom row) and average intervals between targets (**d**, **g**), hits (**e**, **h**), or misses (**f**, **i**) while these fluctuations were measured.

of stronger entrainment and lower α-power alternated with periods of weaker entrainment and higher α-power. If such periods are indeed indicative of different attentional processing modes, they should also be related to target detection. In other words, α-power (or entrainment) fluctuations prior to detected targets should be in opposite phase as compared to missed ones. Figure 5a, c depicts the average time courses of α-power and entrainment fluctuations around hits and misses (bandpass filtered around the individual peak frequencies around 0.07 Hz). We computed the instantaneous phase of these time courses on an individual level. We observed a systematic clustering of phase differences between hits and misses for both α-power (Rayleigh test: $p = 0.009$, $z = 4.61$), and entrainment fluctuations (Rayleigh test: $p = 0.049$, $z = 2.07$) in the time period before a target occurred (−14 s to −2.5 s). Importantly, the mean angle of this difference significantly differed from 0 (circular one-sample test against angle of zero; alpha: $p < 0.01$, $M_{angle} = -2.87$, $CI_{99} = 2.42$, −1.88; entrainment: $p < 0.05$, $M_{angle} = 2.44$ rad, $CI_{95} = 1.52$, −2.93). The $CI_{99}$ of this difference included ± π (Fig. 5b, e), indicating an anti-phase relation. The observed phase relations appeared to be relatively stable across the pre-stimulus time period (Fig. 5c, f). Together, we found that the hypothesized markers of attentional processing (entrainment vs. α-power) differ depending on whether an auditory target was detected or not, as expected from modes of external and internal attention, respectively.

## Discussion

Variations in performance are a prominent feature of sustained attention. In the current study, our marker of sustained attention to speech—neural entrainment—exhibited slow fluctuations with an inherently rhythmic component (Fig. 2a). Importantly, these fluctuations were opposite to those in α-oscillations (Fig. 2f), commonly assumed to reflect suppressed sensory input[13,14] and therefore indicative of an opposite mode of "internal" attention. In addition, neural signatures of attentional mode differed depending on whether a target was detected or not (Fig. 5). Our results therefore demonstrate that lapses in (external) attention occur rhythmically, even when presented with a stimulus that requires sustained attention for successful comprehension. Moreover, these fluctuations occurred at time scales (~14 s) that are very similar to those observed in non-human primates[10]. Thus, we might have tapped into a general property of sustained attention that is conserved across species. The questions whether attention to sensory input always includes regular lapses, and which experimental manipulations can make attention more "sustained" need to be addressed in future work.

Our results indicate that fluctuations in attentional modes may emerge from interactions of distinct brain networks. Defined as neural activity aligned to a stimulus rhythm[19,20,22], we here used neural entrainment as a marker for sensory processing, and therefore external attention. We used rhythmic speech as the entraining stimulus, given its role in human communication and ability to entrain endogenous brain oscillations[19,20,22,23,40]. In

**Fig. 4 | Source-level analyses of α-power and entrainment fluctuations.** Neural activity index (NAI) of rhythmic fluctuations in entrainment (**a**) and α-power (**b**). **c** Coupling between α-power and entrainment fluctuations across 360 cortical ROIs. Each circle indicates the location of an ROI that belongs to a cluster with significant coupling. The size of the circle indicates the number of ROIs significantly coupled to each ROI. Lines indicate coupling between an ROI's entrainment fluctuations and another ROI's α-power fluctuations. Darker shades of red indicate stronger coupling. We found three clusters with significant coupling—1: parietal (α) to right temporal (entrainment), 2: right frontal (α) to left temporal (entrainment), 3: parietal (α) to left frontal (entrainment). **d** Overview of cortical structures, extracted from the HCP-MMP1 atlas, that contain ROIs with coupling between α-power and entrainment fluctuations. Thicker, darker lines indicate more connections between ROIs in two given structures. Font color indicates if a structure shows (predominantly) entrainment (violet) or α-power coupling. **e** Distribution of phase differences between α-power and entrainment fluctuations from all ROIs that are part of the significant clusters.

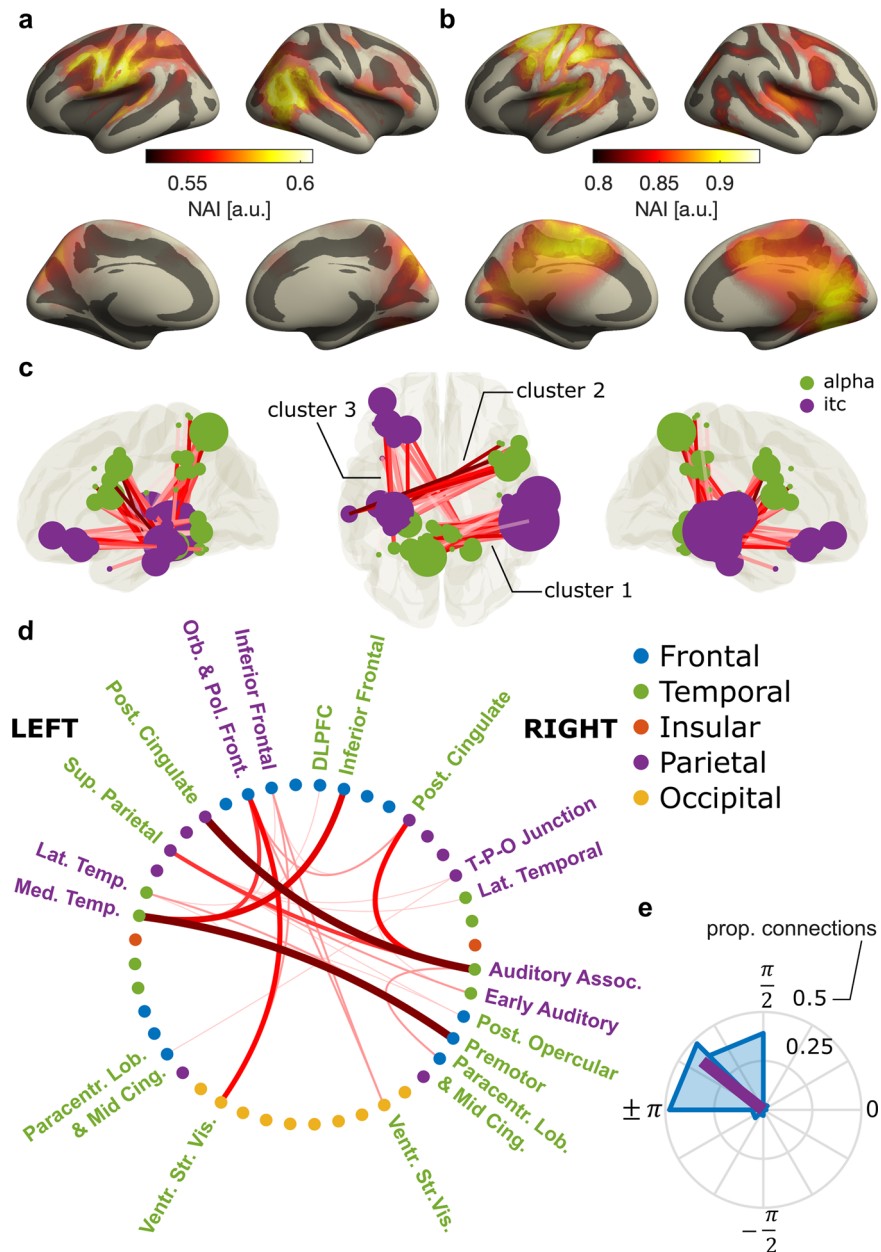

line with this approach, the network exhibiting slow, rhythmic fluctuations in entrainment entailed auditory and inferior frontal regions that typically entrain to auditory stimuli and speech in particular[40,41], and are involved in phonological and semantic processing[42].

Our results imply that, in moments with reduced external attention and corresponding decrease in neural entrainment, α-power increases in a network including Superior Parietal Lobe, Posterior Cingulate Cortex, and right IFG. Many of these regions have been associated with dorsal (and ventral) fronto-parietal attention networks[37,38,43]. These networks have previously been suggested to support attention in different modalities, although the exact regions involved may not overlap completely[44,45]. However, they have been predominantly studied in vision and evidence for their involvement in auditory, and particularly in sustained attention is scarce. Parts of Posterior Cingulate Cortex are also implicated in the default mode network[39]. Activity in this network has been associated with internal attention and reduced processing of external stimuli, resulting in attentional lapses[1,2]. Importantly, attention and default mode networks are typically localized with functional magnetic resonance imaging (fMRI), which offers high spatial resolution, but cannot resolve the role of brain oscillations in

the involved regions. In contrast, previous research has linked α-oscillations to inhibition of sensory processing[13,14,46,47] and attentional (de-) selection[15–18,48]. In the context of sustained attention, high levels of α-oscillations are associated with attentional lapses[10,11,49] and periods of mind wandering[50]. However, these fields of research mostly exist in isolation, and it remained unclear to what extent brain networks found in fMRI and those that give rise to α-oscillation effects in attention overlap. Although causality needs to be tested in follow-up work, our results suggest that dorsal attention and default mode networks may exercise rhythmic top-down control of sensory processing via α-oscillations, leading to slow, regular changes between internal and external attention.

While previous research has investigated the role of α-oscillations and neural entrainment to speech, only few works have considered them in conjunction[24–26]. Although two of these studies indicate some link between α-oscillations and neural entrainment[24,26], other work suggested that they are independent measures of neural activity[25], in apparent conflict with the current findings. However, most of this earlier work focused on fluctuations in α-power lateralization over time and consequently, on spatial rather than sustained attention. The "lateralization index" employed in these studies

**Table 1 | Regions according to HCP-MMP1 atlas showing auditory entrainment or α-power coupling**

| Cortical structure | Measure | ROIs | Hemisphere |
|---|---|---|---|
| Medial temporal | ITC | Hippocampus, Para-Hippocampal Area 3, Area TF, Para-Hippocampal Area 2 | L |
| Lateral temporal | ITC | Area TE1 Middle, Area TG dorsal | L |
| Inferior frontal | ITC | Area 47l (47 lateral), Area anterior 47r, | L |
| Orbital and polar frontal | ITC | Area 13l, Area 47s, Area 47m | L |
| Posterior opercular | ITC | Area OP4-PV | R |
| Early auditory | ITC | Para-Belt Complex, Medial Belt Complex | R |
| Auditory association | ITC | Area STSd posterior, Area STSv posterior, Auditory 4 Complex | R |
| Temporo-parieto-occipital junction | ITC | Area Temporo-Parieto-Occipital Junction 1 | R |
| Lateral temporal | ITC | Area TE1 Middle | R |
| Posterior cingulate | alpha | Retro-Splenial Complex, Pre-Cuneus Visual Area, Area dorsal 23 a + b, Area 31p ventral, Area 31pd, Area 31a | L |
| Paracentral lobular & mid cingulate | alpha | Area 5L | L |
| Superior parietal | alpha | Lateral Area 7A, Medial Area 7A | L |
| Ventral stream visual | alpha | Fusiform Face Complex, Ventro-Medial Visual Area 1, Ventral Visual Complex | L |
| Medial temporal | alpha | Pre-Subiculum, Para-Hippocampal Area 1 | L |
| Premotor | alpha | Premotor Eye Field, Area 55b, Rostral Area 6 | R |
| Posterior cingulate | alpha | Pre-Cuneus Visual Area, Area dorsal 23 a + b, Area 31a, Pro-Striate Area | R |
| Paracentral lobular & mid cingulate | alpha | Area 5m, Area 5L | R |
| Dorsolateral prefrontal | alpha | Area 8C | R |
| Inferior frontal | alpha | Area 44, Area IFJa | R |
| Posterior opercular | alpha | Area OP2-3-VS | R |
| Ventral stream visual | alpha | Ventro-Medial Visual Area 1 | R |

Overview of cortical structures with ROIs that show auditory entrainment or α-power fluctuations significantly coupled to other ROIs. Labels are according to the HCP-MMP1 atlas. Measure indicates if ROIs show ITC coupled to α-power in other ROIs (ITC) or α-power coupled to ITC in other ROIs (alpha). Hemisphere labels correspond to: L = left, R = right.

might have removed slow fluctuations in α-power that occur during sustained attention. Moreover, the trial-based experimental design used is—due to frequent interruptions or changes in stimulus presentation—likely inadequate to capture slow changes in attention that we hypothesized to occur during repetitive sensory stimulation. Accordingly, the constant 3 Hz rhythm in our experimental design might have been crucial to discover the slow anti-phasic fluctuations reported here. Indeed, research in the visual domain that did employ continuous reaction tasks[11,46,47] used α-power to predict lapses in attention up to 20 s before they occur. Interestingly, although not explored by the authors, α-power time courses seemed to contain a slow rhythmic component similar to the one observed here (cf. Fig. 5 in ref. [11]). Nevertheless, the fact that α-oscillations and neural entrainment fluctuate in slow anti-synchrony during sustained attention but not necessarily on a shorter time scale is an interesting observation that deserves closer examination.

It remains unclear why the observed effects vanish when participants close their eyes. Closing the eyes is known to cause a substantial increase of α-oscillations pre-dominantly in visual areas[29]. It is possible that these enhanced visual α-oscillations overshadow their auditory and parietal counterparts when eyes are closed, such that their coupling to auditory entrainment cannot be traced anymore. Alternatively, eye-closure may cause a fundamental change in the brain's processing mode. Blocking visual input may allow to allocate more cognitive resources to auditory processing, such that rhythmic switching may occur at fundamentally different frequencies or is not necessary at all.

Our results pose important questions about the putative mechanisms driving the remarkable rhythmicity of attentional fluctuations observed in our data and their function. An obvious concern might be that the rhythmicity is inherently driven by the regularity of the stimulus material. However, this seems unlikely. Low-frequency effects caused by the rhythmicity of the stimulus material should follow the principles of synchronization theory, which would predict such effects to occur at precise, predictable subharmonic frequencies[51], while the fluctuations we observe in our data vary across participants. In a control analysis, we did not find any relationship between the rate of targets and the frequency of slow α-power and ITC fluctuations (Fig. 3).

It therefore seems likely that the observed fluctuations are intrinsically driven. Indeed, at the level of short sub-second time-scales it has been repeatedly suggested that perceptual and attentional sampling are inherently rhythmic, fluctuating in the range of theta and α-oscillations[52–59]. It may thus be plausible that rhythmicity in attention can also exists at other time scales. We speculate that regular changes between internal and external attention reflect a protective mechanism to prevent depletion of attentional resources. Fluctuations in attention may allow the system to maintain a higher-level of performance for a longer period of time at the cost of regular periods of reduced sensory processing. Indeed, evidence from non-invasive brain stimulation suggests that enhancing endogenous α-oscillations with electrical stimulation has a stabilizing, rather than a decremental effect on sustained attention[60].

There is an intriguing similarity of timescales between the intrinsic attentional rhythms in our data and the known coupling between α-oscillations and slow rhythmic activity in the respiratory system[61], the heart, and the gut[62]. In particular, the gastric network seems to generate rhythmic activity at similar time scales as those we found here (~0.05 Hz)[62,63], and modulate the amplitude of spontaneous α-oscillations[64]. Future research should address if these similarities in time scales are functionally meaningful. A recent magnetoencephalography (MEG) study supports this notion as it localized gastric-alpha coupling to right IFG and parietal lobe[64], regions that overlap with the attention-related networks reported in our study.

A systematic rhythmicity of attentional fluctuations has important implications for both basic and applied research. Considering rhythmicity may make attentional lapses more predictable and offer a potential target for interventional approaches. For example, transcranial alternating current

**Fig. 5 | α-power and entrainment fluctuations differ between detected and missed targets.**
**a** Average, bandpass filtered α-power fluctuations around hits and misses extracted from significant cluster shown in Fig. 2e. Note that data after −2.5 s can be affected by "smearing" of post-target data and thus cannot be interpreted in light of our hypothesis.
**b** Polar histogram depicts the distribution of phase differences between α-power fluctuations prior to hits and misses, respectively (−14 s to −2.5 s).
**c** Event-related phase difference of α-power fluctuations prior to hits and misses. Thin lines indicate single subject time-courses. Bold line depicts the circular average. Data are only shown for the time period used for statistical analysis. Note that −π = π, suggesting a stable phase opposition between envelopes preceding hits and misses, respectively.
**d–f** Same as (**a–c**), but for ITC. Shaded areas indicate standard error of the mean.

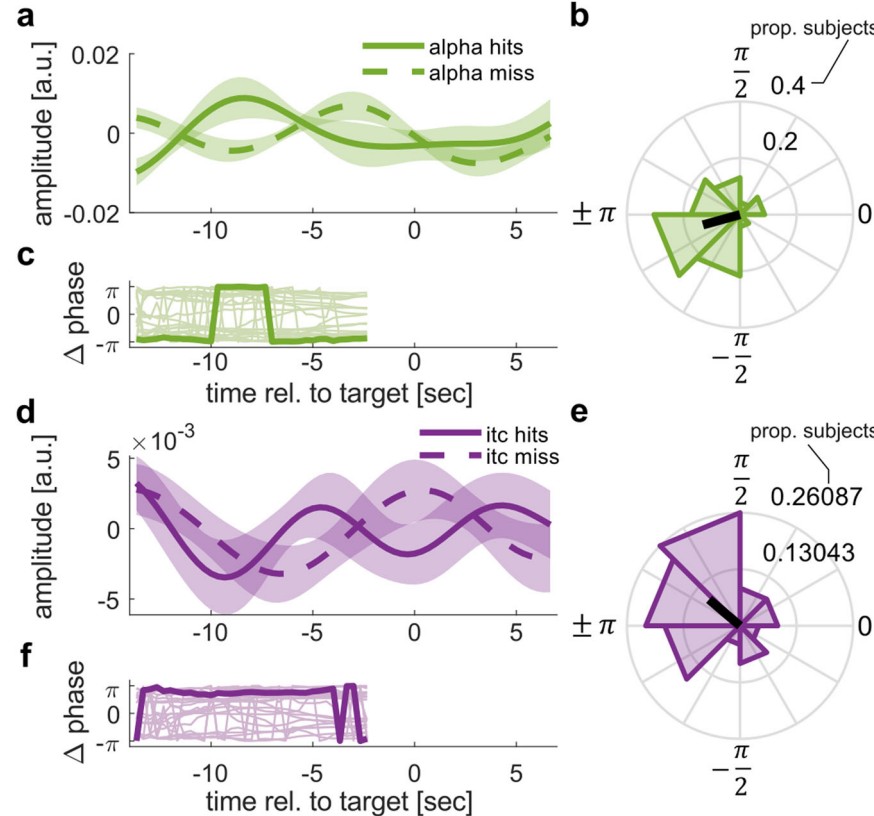

Over the course of six 5-min blocks, participants were instructed to listen to continuous streams of rhythmic, monosyllabic French words and to indicate if they detected deviations from the rhythm via a button press on a standard computer keyboard. In the beginning of each block, they were instructed to either keep their eyes-open and fixated on a white cross at the center of a computer screen, or to keep them closed. Half the blocks were assigned to the eyes-open, and eyes-closed conditions respectively. The order of blocks was randomized to avoid time-on task effects. Participants were familiarized with the task prior to the main experiment. They were shown examples of continuous rhythmic speech trains as well as streams containing violations of the rhythm. Subsequently, they performed a 1-min practice run of the task.

stimulation (tACS) can be used to modulate brain oscillations[65–67] and has been previously shown to stabilize sustained attention when applied in a continuous manner[60]. Considering fluctuations of attentional modes may allow to apply tACS in a state-dependent manner, e.g., to induce shifts in the attentional state by applying stimulation either in the α-frequency range or in synchrony with the external stimulus. Such targeted intervention may offer novel opportunities to improve or steer sustained attention performance in critical systems or in neurological or psychiatric patients suffering from deficits in sustained attention[7,8]. Our source-localization results reveal regions that can be targeted to modulate attentional modes. Before moving to such practical applications, additional research should investigate to what extent the current findings generalize across sensory systems. Compared to other sensory domains with a more static input, audition is special in that information is inherently transient and may thus benefit more from processing principles that take its temporal structure into account[68]. It thus remains to be determined if similar rhythmicity exists when sustained attention is deployed to visual, somatosensory, or cross-modal tasks and if so, at which frequencies rhythmicity emerges, and whether the same attentional networks engage in its regulation. First evidence for similar effects in the visual domain has already been reported in non-human primates[10].

## Materials and methods
### Participants
Twenty-three healthy volunteers (age 22.4 years ± 1.6 years, 15 females) participated in the study. They gave written informed consent prior to joining the experiment and were paid for their participation. The study was approved by the ethics board CPP (Comité de Protection des Personnes) Ouest II Angers (protocol no: CPP 21.01.22.71950 / 2021-A00131-40). All ethical regulations relevant to human research participants were followed.

### Experimental design
Over the course of six 5-min blocks, participants were instructed to listen to continuous streams of rhythmic, monosyllabic French words and to

### Apparatus and stimuli
Original recordings consisted of a set of 474 monosyllabic French words, spoken to a metronome at a rate of 2 Hz by a male, native French speaker. This approach aligned perceptual centers (p-centers)[69] of the words to the metronome beat and resulted in perceptually very rhythmic speech (see Zoefel et al.[70] for a detailed description of stimuli and task). Stimuli were then time-compressed to 3 Hz using the pitch-synchronous overlap and add (PSOLA) algorithm implemented in the Praat software package, and the metronome beat was made inaudible to participants. Intelligibility of the speech recordings was degraded by applying 16-channel noise-vocoding[71]. 16-channel noise-vocoded speech is not as easy to understand as clear speech, but is still clearly intelligible[72]. Individual, noise-vocoded words were then concatenated into a continuous sound stream of 5-min length (i.e., totaling 900 words). The order of words was randomized with a constraint such that every word from the stimulus set had occurred before it can be repeated. Across all blocks, 150 target words that deviated from the 3 Hz stimulus rate by 80 ms (50% presented early, 50% late) were embedded into the rhythmic speech stream (25 targets per block, 75 targets per eyes-open/eyes-closed condition). Participants were asked to indicate violations of the rhythm by pressing the space bar on a standard computer keyboard.

The experiment was carried out in a dimly-lit recording chamber, separated from the experimenter. Audio signals were generated in MATLAB 2019a and streamed to a Fireface UCX (RME Audio,

Heimhausen, Germany) soundcard. The audio stream was presented to participants using a pneumatic In-Ear headphone system (Etymotic Research ER-2, Etymotic Research Inc., USA). This system provides an additional layer of shielding from environmental noise. Experimental instructions were given via a computer screen in the experimental room controlled using Psychtoolbox 3 for MATLAB. During blocks that required participants to keep their eyes open, a white fixation cross was presented on a black background at the center of the screen. The fixation cross was also shown during eyes-closed conditions to keep environmental light conditions constant across blocks.

## EEG

Electroencephalogram was recorded from 64 active electrodes according to the extended international 10-10 system using a BioSemi Active 2 amplifier (BioSemi, Amsterdam, Netherlands). EEG signals were recorded at a rate of 2048 Hz and digitally stored on a hard drive using ActiView v9.02 Software (BioSemi, Amsterdam, Netherlands). Electrodes were mounted in an elastic cap and connected to participants' scalps via a conductive gel (Signa Gel, Parker Laboratories Inc., Fairfield, NJ, USA). Signal offsets of the system were kept below 50 µV.

## EEG processing

EEG analyses were performed in MATLAB 2019b using the fieldtrip toolbox[73]. Data was re-referenced to common average, resampled to 256 Hz and filtered between 1 Hz and 40 Hz using a two-pass, 4th order, zero-phase Butterworth filter. An independent component analysis was performed to project out artifacts related to eye-blinks, movements, heart-beat or muscular activity.

Signals were then epoched into consecutive, overlapping segments centered around the p-center (the part of the word that was centered on the metronome beat) of each word (±1 s). Each segment was 2 s long and therefore comprised of seven p-centers (Fig. 1). The use of segments allowed us to extract time-resolved measures of α-oscillations and neural entrainment.

A Fast-Fourier Transform (FFT, Hanning window, 2 s zero padding) was applied on each of the segments. The resulting complex Fourier coefficients were used to extract power in the α-band (8–12 Hz) as well as inter-trial coherence (ITC) at the stimulus rate (3 Hz). In line with previous work[10,23], we used ITC to quantify neural entrainment to speech, i.e., neural activity aligned to the 3 Hz rhythm. ITC quantifies phase consistency across trials (here: segments). ITC was computed in sliding windows comprising 15 segments (step size: 1 segment). Note that, due to the overlap between successive segments, this window is 5 s long. We used the following equation to compute ITC in each time window:

$$ITC(f) = \left| \frac{1}{N} \sum_{n=1}^{N} e^{-i(\varphi(f,n))} \right|$$

where $\varphi(f,n)$ is the phase in segment $n$ at frequency $f$. $N$ corresponds to the number of segments in the window. $f$ was therefore set to 3 Hz and $N$ was set to 15. Within the same windows we averaged power spectra across segments to ensure consistent temporal smoothing in both measures. Power and ITC spectra were visually inspected to ensure good ratio between spectral peaks and the "noise" at the surrounding frequencies as well as to ensure plausible topographies.

This approach yielded neural measures as a function of time: One α-power, and one ITC value per time window. We then used these time-resolved measures to extract their fluctuations over time. ITC and α-power time-series were first z-transformed to ensure comparable amplitudes. They were then divided into 100 s segments with 90% overlap. This resulted in a total of 60 segments across the 3 blocks per condition (eyes-open vs. eyes-closed). Finally, the segments were submitted to another FFT (Hanning window, 400 s zero padding). Length of the segments and padding were chosen to ensure sufficient spectral resolution below 0.2 Hz as well as a

sufficient number of phase estimates to quantify if there is systematic coupling between signals. Importantly, a 90% overlap results in an effective step size of 10 s in our case. Given our aim to analyze fluctuations at rates below 0.2 Hz (i.e., slower than 5 s), a 10 s step size is sufficient to include 0.5–2 new cycles of such slow fluctuations. To rule out that these step and window size parameters affect our results we repeated the analysis using shorter segments (50 s with 90% overlap, i.e., 5 s time-steps). Overall, we obtained very similar results compared to the main analysis.

The obtained low-frequency spectra were corrected for arrhythmic ("1/f") activity using the fooof algorithm[74] as shipped with the fieldtrip toolbox. This step helped improve the identification of peaks in the spectrum and revealed prominent spectral peaks in the α-power and ITC time-courses. On the group level, these peaks were close to 0.07 Hz for both α-power and ITC (Fig. 2a). We next identified individual peak frequencies of α-power and ITC fluctuations. To do so, we selected the peak frequency that was closest to 0.07 Hz rhythm for individual participants (in a range of 0.04 Hz to 0.1 Hz). In accordance with Lakatos et al.[10], we then used the individual peak frequency of the α-power envelope to extract the phase of α-power and ITC fluctuations in each 100 s segment, and computed their phase difference. Phase differences were subsequently averaged across segments, separately for each subject and condition. To rule out that coupling between α-power and entrainment are driven by entrainment effects at harmonic frequencies in the α-band (9 Hz and 12 Hz), we repeated the analysis for the coupling between fluctuations in 3 Hz ITC and ITC at 9 Hz and 12 Hz. The results of this analysis are presented in Supplementary Fig. S3.

To investigate how α-power and ITC fluctuations relate to the detection of target stimuli, we filtered the corresponding envelopes around the individual α-power envelope peak frequency (±0.02 Hz), identified in the previous step, using a causal, 6th order, one-pass Butterworth filter. The causal filter was chosen to avoid contamination of pre-stimulus activity with stimulus related changes in brain activity. We epoched the signals from −14 s to +7 s around target stimuli. Using a Hilbert transform, we then extracted instantaneous phase angles over time and averaged them across trials, separately for ITC and α-power fluctuations around hits and misses, respectively. Subsequently, for both α-power and ITC fluctuations, we computed the time-resolved phase difference between hits and misses in the interval before target onset (−14 s to −2.5 s). The interval ends 2.5 s prior to the onset of target stimuli to avoid including target-evoked brain responses, which can smear into the interval due to the symmetric 5 s window (described above) that was used to compute ITC values and to smoothen α-power trajectories. The analysis was restricted to channels from significant clusters revealed in previous analyses (Fig. 2d, e and Supplementary Table 1).

## Source analysis

We applied dynamic imaging of coherent sources (DICS) beamforming[31] at individual peak frequencies of the slow fluctuations in α-power and ITC. Leadfields were constructed from the standard boundary element model of the fieldtrip toolbox, and standardized electrode locations were defined in MNI space. We projected the complex Fourier coefficients from each 100 s segment onto a surface grid with 20,484 source locations. From these complex coefficients, we obtained a neural activity index by dividing oscillatory power by an estimate of the noise bias. To quantify coupling between α-power and ITC fluctuations for different combinations of brain regions, we computed the phase angle of these two neural measures for each segment, source location, and subject. We then applied a parcellation into 360 regions of interest (ROIs) according to the HCP-MMP1 atlas of the human connectome project[32]. To this end, we computed the average phase across all source locations within an ROI. We then computed the average phase across the 60 segments for each ROI and subject, separately for α-power and ITC fluctuations, and tested for consistent phase relations between the two (our measure of connectivity) across subjects, as detailed below. The parcellation of source-level data reduces the number of channel-channel comparisons necessary and provides meaningful anatomical labels,

thereby improving interpretability and reducing computational demands. It also mitigates localization errors arising from the use of standard head models and electrode locations for all participants, as brain activity is integrated over larger areas of the brain.

## Behavioral analysis

Average reaction times, as well as the proportion of hits, misses and false alarms were computed for each condition (eyes-open vs. eyes-closed). A target word was considered a hit, if a button was pressed within 2 s after stimulus onset, otherwise it was considered a miss. Button presses occurring outside these response intervals were considered false-alarms. Hit rates were computed by dividing the number of responses to targets by the number of targets (75 per condition), false alarm rates were computed by dividing responses outside of response intervals by the number of standards (2475 per condition).

## Statistical analyses and reproducibility

Statistical analyses were performed in Matlab 2019b using the circular statistics toolbox[75] in combination with functions in fieldtrip for massive-multivariate permutation statistics[73,76]. All statistical analysis were conducted on $n = 23$ participants.

To assess whether there is a significant coupling between α-power and entrainment fluctuations (i.e., envelopes), we tested if their phase difference shows a systematic clustering across subjects (i.e., differs from a uniform distribution). To this end, we first computed the average phase difference between α-power and ITC envelopes (separately for each channel and subject as detailed in EEG Processing) and subjected them to Rayleigh's test for uniformity of circular data. This test yields a z-statistic (one per channel) which is high if these phase differences are non-uniformly distributed across subjects. We next tested whether there is a cluster of channels with such a non-uniform distribution. We randomly shuffled α-power and ITC segments within subjects 10,000 times, re-computed the average phase difference between the two signals per-subject and the corresponding Rayleigh's statistic on the group level, yielding 10,000 z-statistics per channel. Finally, we compared actual data with shuffled ones to obtain group-level $p$ values for channel clusters, using Monte Carlo estimates[76].

A similar statistical approach was used to test for significant coupling of α-power and ITC fluctuations in the parcellated source-level data. In contrast to the sensor-level analysis, the test was run for all possible combinations of ROIs. To keep the resulting network structures sparse, we employed a more conservative α-level of 0.01 for this test.

A significantly non-uniform distribution of phase differences between α-power and ITC envelopes indicates that the two are coupled but does not tell us anything about their phase relation. If this phase relation is close to 0, this would speak against α-oscillations and entrainment reflecting opposing processing modes. We therefore tested the phase relation between the two deviates from zero within an identified channel cluster, using the circ_mtest function of the circular statistics toolbox. The function tests if a given angle lies within a given confidence interval (e.g., 95% or 99%) of a phase distribution. To estimate a $p$ value, we performed the test against different significance levels ($\alpha < 0.05$ and $\alpha < 0.01$) and report the lowest significant α-level for each comparison (i.e., $p < 0.05$ or $p < 0.01$), along with the circular average of the underlying phase distribution and the 95% or 99% circular confidence intervals (CI). Although a non-zero-phase relation between α-oscillations and entrainment might already indicate different modes of processing, our hypothesis was more explicit in that it assumes an opposing (i.e., anti-phase) relation between the two. To evaluate this hypothesis, we also report if the CIs cover $\pm\pi$.

Comparisons of peak frequencies, hit and false alarm rates and reaction times were performed in Matlab 2019b using dependent samples $t$-tests.

## Reporting summary

Further information on research design is available in the Nature Portfolio Reporting Summary linked to this article.

## Data availability

The underlying anonymized data is available in minimally processed form (down sampled, filtered, artifacts suppressed with ICA) via the open science framework (https://osf.io/64ycj/). Due to the file size, the unprocessed raw data can be obtained upon reasonable request from the corresponding authors. Source data underlying the graphs in the manuscript are provided as Supplementary Data.

## Code availability

The MATLAB code used to process and analyze the data is available via the same open science framework repository (https://osf.io/64ycj/).

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

## Acknowledgements

This research was supported by a grant from the Agence Nationale de la Recherche (grant number ANR-21-CE37-0002) awarded to B.Z.

## Author contributions

Florian H. Kasten: conceptualization, methodology, software, data collection, formal analysis, investigation, writing—original draft, visualization; Quentin Busson: data collection, software, writing—review & editing; Benedikt Zoefel: conceptualization, writing—review & editing, supervision, funding acquisition.

## Competing interests

The authors declare no competing interests.
