## [Transparent Peer Review file · Communications Biology]

Opposing neural processing modes alternate rhythmically during sustained auditory attention

Corresponding Author: Dr Florian Kasten

Figures originally included in the author's rebuttal have been redacted from this file.

Version 0:

Reviewer comments:

Reviewer #1

(Remarks to the Author)

The manuscript by Kasten and colleagues describes a human EEG study that examined fluctuations in alpha activity and neural entrainment. They found that the two are in anti-phase such that high alpha is related to low entrainment and visa versa. These fluctuations were on the time scale of 0.07 Hz, reminiscent of the animal literature. Although the methods are sound, there are some issues that prevent me recommending this paper for publication in its current form. Although I really like the idea that alpha and entrainment are in anti-phase, this makes sense and is appears to be logical, previous literature on this topic would suggest that this not always the case. Often times, alpha and auditory entrainment are reported separately. Three papers that come to mind are Hauswald et al., 2020, Tune et al., 2021 and Kerlin et al., 2010. The first two studies found relative independence for alpha and entrainment. Hauswald for modelling behavior. These discrepancies in the literature should be mentioned. One can argue, that perhaps sliding the time course method employed in the current paper was able to resolve the above issues if they are "washed out" by averaging over the entire time period of the condition. However, the Tune paper looked at trial-to-trial alpha and entrainment and found independence. The Kerlin study would suggest that they are related. Nonetheless, mentioning these are important and help clarify the literature given the apparent discrepancy. There is a whole literature on pre-stimulus alpha predicting behaviour on "go no-go" experiments (although cited, the authors do not explore). The other main issue that is perhaps more worrisome is that the effects described by Kasten et al., are artifacts arising motor preparation and execution. Motor-related alpha is a necessary "evil" and we cannot rule out these factors. Especially worrisome is that the deviant is occurring at 0.08 Hz (25 targets in a 300 sec block leads to 0.08 Hz). This is very close to 0.07 Hz rhythm described. In Figure 3, the alpha has a dominant generator near the left motor cortex (right-handed participants?). Granted that the hit rate was 44%, this could be a washed-out effect when averaged across hits and misses. Was this activity greater when averaged separately for hits vs. misses? If so, then this could suggest that a motor origin and not a fluctuation reminiscent of animal models and internal modes of attention. I think this needs to be addressed and ruled out in order to make claims of inherent intrinsic modulations of alpha.

Reviewer #2

(Remarks to the Author)

In the manuscript COMMSBIO-23-4855-T entitled " Opposing neural processing modes alternate rhythmically during sustained auditory attention" by Dr. Florian H. Kasten and colleagues, the authors describe periodic fluctuations between modes of neural entrainment to speech and spontaneous α -oscillations in two distinct brain networks during detection of auditory targets. The authors hypothesize in the manuscript that this might be a conserved attentional mechanism being critical during sustained attention to sensory information. Overall, the manuscript is well written and of interest to the reader of Communications Biology. Applied methods are appropriate and study outcomes are sound and well presented. Still, there are some points that should be addressed by the authors before publication. These can be summarized as follows: Study participants are all in the same narrow age range of 20 to 24 years. Can results obtained for this young age group also be expected for older subjects? Were differences observed between female and male individuals? The authors reported that 14 out of 23 participants show a significant coupling within the identified cluster. Is there something known about the difference between study participants where significant coupling occurred in comparison to the

others where this not occurred, e.g. concerning other EEG features or descriptive data?

EEG processing and source analysis as described by the authors in the method section is complex depending on multiple processing steps. How is data quality ensured and controlled during conduct of the different processing steps? Does there exist certain thresholds or background noise levels? If there are such quality parameters or indicators, these should be mentioned by the authors.

In respect to the differences between eyes-open vs. eyes-closed, how can visual factors be excluded as root cause for the observed differences, e.g. additional sensory stimulus to steadily fix the cross during conducting the listening and reacting tasks?

It is outlined by the authors that their observations might be conserved across species, since similar results were reported previously in two macaque monkeys. Is there further evidence known by the authors that support the possibility of a conserved mechanism? Is here something known to the authors, where this mechanism might originate?

What is not clear from reading the manuscript, is the question if the periodicity of ~ 0.07 Hz is limited to auditory targets or if the same frequency can be expected for visual and other persistent attentional stimuli. The authors should be clearer on this. Is there something known about how variances in the experimental setup have impact on the study outcome? The authors should describe in a bit more detail, what are the experimental prerequisites that can trigger these periodic oscillations. Is there something like a transition point?

Author Rebuttal letter:

Kasten et al. 2024 Response to reviews

Response to Reviews

We would like to thank both reviewers for their positive feedback as well as the thorough and thoughtful comments on the manuscript. In the following we provide our point-by-point responses to the issues raised. Original reviewer comments are provided in **Bold font**, excerpts from the revised manuscript are in *italics*, our responses to the comments are in normal font. Changes in the main manuscript are indicated in yellow.

Reviewer #1 (Remarks to the Author):

The manuscript by Kasten and colleagues describes a human EEG study that examined fluctuations in alpha activity and neural entrainment. They found that the two are in anti-phase such that high alpha is related to low entrainment and *visa versa*. These fluctuations were on the time scale of 0.07 Hz, reminiscent of the animal literature. Although the methods are sound, there are some issues that prevent me recommending this paper for publication in its current form.

1. Although I really like the idea that alpha and entrainment are in anti-phase, this makes sense and it appears to be logical, previous literature on this topic would suggest that this not always the case. Often times, alpha and auditory entrainment are reported separately. Three papers that come to mind are Hauswald et al., 2020, Tune et al., 2021 and Kerlin et al., 2010. The first two studies found relative independence for alpha and entrainment. Hauswald for modelling behavior. These discrepancies in the literature should be mentioned. One can argue, that perhaps sliding the time course method employed in the current paper was able to resolve the above issues if they are “washed out” by averaging over the entire time period of the condition. However, the Tune paper looked at trial-to-trial alpha and entrainment and found independence. The Kerlin study would suggest that they are related. Nonetheless, mentioning these are important and help clarify the literature given the apparent discrepancy.

Response: We thank the reviewer for pointing us towards this interesting literature. We have now included a more in-depth discussion of the papers particularly in terms of similarities and differences to our current work.

While all three studies investigated the link between neural entrainment to speech and alpha oscillations, they do so with large methodological and conceptual differences. The most critical one, in our opinion, is the trial-based experimental design in these studies. Our focus on slow fluctuations in sustained attention requires constant stimulation without frequent interruptions, which was ensured by the stable 3-Hz rhythm, but which is not possible in trial-based designs. If pauses or changes in experimental conditions (e.g., intelligibility in the Hauswald study) between trials interrupted sustained attention, then this explains why the slow anti-phasic fluctuations in alpha power and entrainment were not detectable in this earlier work. Another difference is that some of these studies investigate spatial rather than sustained attention and computed an alpha power lateralization index that is likely to cancel out variance due to sustained attention. To clarify these points, we have included the following in the revised manuscript:

In Ln 48:

[...] Each of these opposing attentional modes might have its own signature, including specific patterns of neural connectivity^{1,2} and neural oscillations^{10–12}. Prominently, α -oscillations (~ 10 -Hz) have been linked to suppression of sensory information^{13,14} and attentional (de-)selection^{15–18}, and might therefore correspond to a state in which input is prone to be ignored. When listeners attend to rhythmic sequences, neural activity synchronizes to the stimulus rhythm, an effect often termed neural entrainment^{19–21}. As a potential counterpart to α -oscillations, neural entrainment is therefore a prime example for a marker of active stimulus processing^{20,22,23}. However, neural entrainment and α -oscillations are often studied in isolation, and the few efforts to relate them^{24–26} have used trial-based paradigms that, due to their frequent interruptions of stimulus presentation, cannot capture fluctuations in sustained attention. One of the rare studies that did consider both of these neural markers during sustained attention measured them in the auditory cortex of macaque monkeys¹⁰. This work supported the notion that internal attention is characterized by strong α -oscillations and reduced sensitivity to external information¹⁰.

1

Kasten et al. 2024 Response to reviews

And in Ln 314:

While previous research has investigated the role of α -oscillations and neural entrainment to speech, only few works have considered them in conjunction^{24–26}. Although two of these studies indicate some link between α -oscillations and neural entrainment^{24,26}, other work suggested that they are independent measures of neural activity²⁵, in apparent conflict with the current findings. However, most of this earlier work focused on fluctuations in α -power lateralization over time and consequently, on spatial rather than sustained attention. The “lateralization index” employed in these studies might have removed slow fluctuations in α -power that occur during sustained attention. Moreover, the trial-based experimental design used is - due to frequent interruptions or changes in stimulus presentation - likely inadequate to capture slow changes in attention that we hypothesized to occur during repetitive sensory stimulation. Accordingly, the constant 3-Hz rhythm in our experimental design might have been crucial to discover the slow anti-phasic fluctuations reported here. Indeed, research in the visual domain that did employ continuous reaction tasks^{11,46,47} used α -power to predict lapses in attention up to 20 seconds before they occur. Interestingly, although not explored by the authors, α -power time courses seemed to contain a slow rhythmic component similar to the one observed here (cf. Fig. 5 in Ref¹¹). Nevertheless, the fact that α -oscillations and neural entrainment fluctuate in slow anti-synchrony during sustained attention but not necessarily on a shorter time scale is an interesting observation that deserves closer examination.

2. There is a whole literature on pre-stimulus alpha predicting behaviour on “go no-go” experiments (although cited, the authors do not explore).

Response: We thank the reviewer for this comment. We agree that a more detailed discussion of a potential link between trial-to-trial variations in task performance and α -power was warranted in our manuscript. As most related work that was cited in our original manuscript did not employ “go no-go” tasks, we are unsure which studies the reviewer refers to. Nevertheless, we have extended the relevant parts of the discussion, emphasizing analytical differences to our approach, and hope that these changes successfully address the reviewer’s concern. As the paragraph concerned is identical to the one reproduced above, we refer to the first point for changes in the manuscript.

3. The other main issue that is perhaps more worrisome is that the effects described by Kasten et al., are artifacts arising motor preparation and execution. Motor-related alpha is a necessary “evil” and we cannot rule out these factors. Especially worrisome is that the deviant is occurring at 0.08 Hz (25 targets in a 300 sec block leads to 0.08 Hz). This is very close to 0.07 Hz rhythm described. In Figure 3, the alpha has a dominant generator near the left motor cortex (right-handed participants?). Granted that the hit rate was 44%, this could be a washed-out effect when averaged across hits and misses. Was this activity greater when averaged separately for hits vs. misses? If so, then this could suggest that a motor origin and not a fluctuation reminiscent of animal models and internal modes of attention. I think this needs to be addressed and ruled out in order to make claims of inherent intrinsic modulations of alpha.

Response: We agree with the reviewer that motor-related “artefacts” are a concern that needed to be ruled out. It is not possible to analyze slow fluctuations in alpha power (or ITC) for hits and misses separately, as this would require an integration over long time periods that typically contain both. However, we addressed the issue by characterizing the regularity of our targets and participants’ responses to them. As shown in the new Figure 3 (reproduced below), none of the distributions of intervals between targets (a), hits (b) or misses (c) shows a bias for ~ 14 s (corresponding to a 0.07 Hz rhythm). In addition, individual frequencies of alpha power and ITC fluctuations are not correlated with the average intervals between targets (d,g), hits (e,h) or misses (f,i) while these fluctuations were measured. It therefore seems unlikely that our results can be explained by the timing of our target stimuli. We have added a

detailed description of the control analysis to the revised manuscript:

Ln 200:

2.5 Slow rhythmic fluctuations cannot be explained by stimulus properties

Although time intervals between targets were selected from a wide range and therefore irregular, their average (25 targets in 300 seconds) resembled the period of the slow neural fluctuations reported. To

2

Kasten et al. 2024 Response to reviews

[...]

Ln 344:

In a control analysis, we did not find any relationship between the rate of targets and the frequency of slow α -power and ITC fluctuations (Fig. 3).

We would also like to point out that results from our network analysis (now Fig. 4c,d) emphasized parietal and temporal cortices (and IFG) as origins of the coupling between α -power and ITC fluctuations, but did not reveal a strong involvement of motor regions.

3

Kasten et al. 2024 Response to reviews

Reviewer #2 (Remarks to the Author):

In the manuscript COMMSBIO-23-4855-T entitled "Opposing neural processing modes alternate rhythmically during sustained auditory attention" by Dr. Florian H. Kasten and colleagues, the authors describe periodic fluctuations between modes of neural entrainment to speech and spontaneous α -oscillations in two distinct brain networks during detection of auditory targets. The authors hypothesize in the manuscript that this might be a conserved attentional mechanism being critical during sustained attention to sensory information.

Overall, the manuscript is well written and of interest to the reader of Communications Biology. Applied methods are appropriate and study outcomes are sound and well presented. Still, there are some points that should be addressed by the authors before publication. These can be summarized as follows:

1. Study participants are all in the same narrow age range of 20 to 24 years. Can results obtained for this young age group also be expected for older subjects? Were differences observed between female and male individuals?

Response: We thank the reviewer for pointing us to the interesting possibility of an age effect. Indeed, the age range for the current study is relatively narrow. As slow rhythmic fluctuations in attention have not been investigated in human participants thus far, we aimed to test our hypothesis in a sample of young healthy adults. Apart from practical reasons (older subjects are harder to recruit for psychology/neuroscience experiments), this seemed a reasonable first step to validate our hypothesis irrespective of age. In line with the reviewer's comment, it is known that both entrainment to speech and alpha oscillations exhibit changes across the life span (e.g., Henry et al., 2017, Nature Communications). It is therefore possible that the periodic fluctuations in entrainment and alpha oscillations also change with age, and possibly account for certain aspects of cognitive or perceptual decline. Although these are exciting questions to test, they went beyond the scope of the current work. However, we acknowledged the possibility of an age effect in the revised text.

As part of the responder/non-responder analysis (described below) we analyzed differences in alpha-entrainment coupling between male and female participants. We did not find statistical differences between the two groups. However, the number of observations per group is relatively small and may not be sufficient to resolve such differences.

Please refer to the next point for corresponding changes to the manuscript.

2. The authors reported that 14 out of 23 participants show a significant coupling within the identified cluster. Is there something known about the difference between study participants

where significant coupling occurred in comparison to the others where this not occurred, e.g. concerning other EEG features or descriptive data?

Response: We thank the reviewer for this question which we addressed in a responder/non-responder analysis. We could not identify statistically reliable differences between these participants, although those with stronger alpha-entrainment coupling also had numerically higher alpha power. It is likely that our dataset lacks statistical power to reliably resolve influencing factors for effective alpha power and ITC coupling. Investigating individual differences requires a sufficient number of responders and non-responders, and therefore a much larger sample size. To justify such an endeavor for future work, we first had to establish the existence of attentional modes in humans, which we achieved with this study.

We added the results of our responder/non-responder analysis to our supplementary materials:

4

Kasten et al. 2024 Response to reviews

Supplementary Note 1

In an exploratory follow-up analysis, we contrasted various oscillatory features in EEG data from participants which exhibit coupling between α -power and entrainment fluctuations (15 “responders” in Supplementary Fig. S1b) with features from those who did not (8 “non-responders” in Supplementary Fig. S1b). Responders exhibited numerically higher power in the α -band (Supplementary Fig. S2a,c) despite near-identical ITC (Supplementary Fig. S2b,d). However, when we compared α -power within the significant cluster that showed coupling to ITC (cf. Fig. 2e), this difference did not reach significance (responder vs non-responder, α -power: $t_{21} = 0.75$, $p = .46$; ITC: $t_{21} = -0.74$, $p = 0.47$; independent samples t-test). There was no reliable difference in α -power and ITC coupling between male and female participants ($t_{21} = 0.95$, $p = .35$, independent samples t-test; Supplementary Fig. S2e,f)

Results are referenced in the main manuscript in Ln 174:

We could not identify reliable differences in oscillatory features or other variables that can distinguish these 15 participants from the others (Supplementary Fig. S2, Supplementary Note 1).

3. EEG processing and source analysis as described by the authors in the method section is complex depending on multiple processing steps. How is data quality ensured and controlled during conduct of the different processing steps? Does there exist certain thresholds or background noise levels? If there are such quality parameters or indicators, these should be mentioned by the authors.

5

Kasten et al. 2024 Response to reviews

Response: We agree with the reviewer on the importance of quality data checks. As we describe in the methods section, we used an ICA decomposition to separate artefacts from brain activity and subsequently remove the former from the EEG signal. This step is necessary for subsequent analyses of continuous EEG data.

We further examined EEG power spectra and topographies for all participants to ensure plausible patterns on a single subject level. We inspected single trial spectra (as time-frequency plots) to verify that no strong outliers impact the data and that clear alpha power and ITC signals are present in all datasets. This step ensured that subsequent analyses were applied to high quality data. Outcome measures that describe the variability across subjects are already provided throughout the manuscript (e.g., Figs. 2, 3, S1, S2). Finally, Fig. 1e,f illustrates that both alpha power and ITC spectra show clear peaks at the expected frequencies and plausible topographies consistent with prior research. Source-level results shown in Fig. 3 are similarly in line with expectations from the literature.

We added a brief description of the visual inspection steps to the methods section of the manuscript:

Ln 452:

Power and ITC spectra were visually inspected to ensure good ratio between spectral peaks and the “noise” at the surrounding frequencies as well as to ensure plausible topographies.

4. In respect to the differences between eyes-open vs. eyes-closed, how can visual factors be excluded as root cause for the observed differences, e.g. additional sensory stimulus to steadily fix the cross during conducting the listening and reacting tasks?

Response: In accordance with standards in EEG research, the experimental setup was kept highly consistent throughout the experiment. Participants were seated in a recording booth separated from the control room by a wall. The booth had no windows and was constantly lit by a battery driven LED lamp. There was no additional sensory stimulation besides the mentioned fixation cross which did not change over time and thus did not introduce fluctuations in neural activity. Participants were instructed to fixate the cross in the center of the screen during the eyes open condition to further minimize variations in visual input due to eye-movements. In response to this comment (and comment 7 below) we added additional details to the methods section of the manuscript:

Ln 414:

The experiment was carried out in a dimly-lit recording chamber, separated from the experimenter. Audio signals were generated in MATLAB 2019a and streamed to a Fireface UCX (RME Audio, Heimbhausen, Germany) soundcard. The audio stream was presented to participants using a pneumatic In-Ear headphone system (Etymotic Research ER-2, Etymotic Research Inc., USA). This system provides an additional layer of shielding from environmental noise. Experimental instructions were given via a computer screen in the experimental room controlled using Psychtoolbox 3 for MATLAB. During blocks that required participants to keep their eyes open, a white fixation cross was presented on a black background at the center of the screen. The fixation cross was also shown during eyes-closed conditions to keep environmental light conditions constant across blocks.

5. It is outlined by the authors that their observations might be conserved across species, since similar results were reported previously in two macaque monkeys. Is there further evidence known by the authors that support the possibility of a conserved mechanism? Is there something known to the authors, where this mechanism might originate?

Response: We are not aware of any studies investigating slow rhythmicity in sustained attention in other species. Although the similarity of our results with previous work in macaque monkey is intriguing, the possibility of an evolutionary conserved mechanism is speculative and warrants more research. We insist on the speculative nature of this point in the revised manuscript.

We require similar speculations to address potential origins of the underlying mechanism. Previous research on sustained attention indicated that alpha oscillations may have a stabilizing effect on sustained attention (Clayton et al., 2019). We propose that a regular switch between internal and external attention can maintain overall performance for a longer period of time, at the cost of short periodic interruptions of sensory processing. Alternatively, the frequency of the slow attentional fluctuations we find shows an intriguing overlap with other bodily rhythms that are known to couple to alpha oscillations, such as gastric rhythms. We mention these hypotheses in the discussion section of the manuscript:

6

Kasten et al. 2024 Response to reviews

Ln 349:

We speculate that regular changes between internal and external attention reflect a protective mechanism to prevent depletion of attentional resources. Fluctuations in attention may allow the system to maintain a higher-level of performance for a longer period of time at the cost of regular periods of reduced sensory processing. Indeed, evidence from non-invasive brain stimulation suggests that enhancing endogenous α -oscillations with electrical stimulation has a stabilizing, rather than a decremental effect on sustained attention⁶⁰.

There is an intriguing similarity of timescales between the intrinsic attentional rhythms in our data and the known coupling between α -oscillations and slow rhythmic activity in the respiratory system⁶¹, the heart, and the gut⁶². In particular, the gastric network seems to generate rhythmic activity at similar time scales as those we found here (~ 0.05 Hz)^{62,63}, and modulate the amplitude of spontaneous α -oscillations⁶⁴. Future research should address if these similarities in time scales are functionally meaningful.

6. What is not clear from reading the manuscript, is the question if the periodicity of ~ 0.07 Hz is limited to auditory targets or if the same frequency can be expected for visual and other persistent attentional stimuli. The authors should be clearer on this.

Response: The original work in macaque monkeys found periodicity in auditory and visual processing (Lakatos et al., 2016). Given its strong environmental rhythms (Ding et al., 2017), we here chose to focus on the auditory modality to test if similar rhythmicity can be found in humans. Due to ethical limits

on data collection, we decided to prioritize statistical power over a high number of conditions and restrict evidence to the auditory domain. Naturally, as the reviewer suggests, the next step is to contrast auditory effects with equivalent ones in vision. Previous findings in macaques described above suggest that these exist. We adapted the manuscript to be clearer on the role of modality for our results:

Ln 79:

In the current study, we recorded electroencephalographic (EEG) data in humans and tested for regular fluctuations in attentional modes while participants paid sustained attention to rhythmic speech sounds. We hypothesized that, similar to the aforementioned findings in non-human primates¹⁰, neural entrainment to speech and spontaneous α -oscillations show rhythmic fluctuations at ultra-slow frequencies close to ~ 0.06 Hz (0.02-Hz - 0.2-Hz) and that these fluctuations show an antagonistic relationship (i.e. are coupled in anti-phase). Given its strong environmental rhythms²⁸, we here chose to focus on the auditory modality. However, our findings may pave the way for investigations into rhythmic attentional fluctuations across sensory modalities.

Ln 374:

[...] additional research should investigate to what extent the current findings generalize across sensory systems. Compared to other sensory domains with a more static input, audition is special in that information is inherently transient and may thus benefit more from processing principles that take its temporal structure into account⁶⁸. It thus remains to be determined if similar rhythmicity exists when sustained attention is deployed to visual, somatosensory, or cross-modal tasks and if so, at which frequencies rhythmicity emerges, and whether the same attentional networks engage in its regulation. First evidence for similar effects in the visual domain has already been reported in non-human primates¹⁰

7. Is there something known about how variances in the experimental setup have impact on the study outcome? The authors should describe in a bit more detail, what are the experimental prerequisites that can trigger these periodic oscillations. Is there something like a transition point?

Response: In line with common practice in experimental psychology and cognitive neuroscience, we took close care to keep experimental conditions as consistent as possible. As explained above, this included a recording chamber with constant light conditions and separated from experimenter, in-ear headphones that reduce environmental noise, and other steps that ensure data quality. We therefore cannot make any statements about variations in the experimental setup beyond the effect of eye closure that is already described in the manuscript. However, we do agree with the reviewer that, in order to facilitate replication of the current findings, our manuscript lacked some methodological details and added these to the manuscript:

7

Kasten et al. 2024 Response to reviews

Ln 414:

The experiment was carried out in a dimly-lit recording chamber, separated from the experimenter. Audio signals were generated in MATLAB 2019a and streamed to a Fireface UCX (RME Audio, Heimshausen, Germany) soundcard. The audio stream was presented to participants using a pneumatic In-Ear headphone system (Etymotic Research ER-2, Etymotic Research Inc., USA). This system provides an additional layer of shielding from environmental noise. Experimental instructions were given via a computer screen in the experimental room controlled using Psychtoolbox 3 for MATLAB. During blocks that required participants to keep their eyes open, a white fixation cross was presented on a black background at the center of the screen. The fixation cross was also shown during eyes-closed conditions to keep environmental light conditions constant across blocks.

In addition to the methodological factors, we now insist on the point that future research needs to reveal experimental variables that can “trigger” the attentional rhythms, as the reviewer suggests:

Ln 285:

The questions whether attention to sensory input always includes regular lapses, and which experimental manipulations can make attention more “sustained” need to be addressed in future work.
